# Model-Based Estimation of Transmission Gear Ratio for Driving Energy Consumption of an EV



Nikolay Hinov [1,*], Plamen Punov [2], Bogdan Gilev [3] and Gergana Vacheva [1]

1 Department of Power Electronics, Technical University of Sofia, Studentski Kompleks, 1756 Sofia, Bulgaria; gergana_vacheva@tu-sofia.bg
2 Automobiles and Transport, Department of Combustion Engines, Technical University of Sofia, Studentski Kompleks, 1756 Sofia, Bulgaria; plamen_punov@tu-sofia.bg
3 Department of Mathematical Modelling and Numerical Method, Technical University of Sofia, Studentski Kompleks, 1756 Sofia, Bulgaria; b_gilev@tu-sofia.bg
* Correspondence: hinov@tu-sofia.bg; Tel.: +359-2-965-2569

**Abstract:** This paper presents a numerical study of the effect of the transmission configuration on the energy consumption of an electric vehicle. The first part of this study is related to a vehicle simulation model that takes into consideration vehicle resistances such as aerodynamic, rolling and inertial resistance as well as the traction force. The model was then validated by means of vehicle acceleration time, from 0 to 100 km/h in the case of a single-speed gearbox. Vehicle power demand and electrical energy consumption were then evaluated over three standardized test cycles: WLTC-Class 3, NEDC and FTP-75. For each cycle, two cases were studied: a single-speed and dual-speed gearbox. Very different power demand was observed between the cycles in terms of maximum and average driving power. The most power-demanding cycle was WLTC, while NEDC was less power demanding. However, the specific driving energy per kilometer was very similar for NEDC and FTP-75, as it respectively accounted to 0.118/0.116 kWh/km and 0.117/0.115 kWh/km. WLTC led to a higher specific consumption of 0.127/0.124 kWh/km. A dual-speed gearbox led to better efficiency, within the range of 1.7% to 2.4%. The higher value was obtained for highly dynamic WLTC.

**Keywords:** electric vehicles; transmission; energy consumption; test cycles; modeling

## 1. Introduction

Recently, a decrease in the availability of the fuel has led to the basic motivation for the creation and studying of electric and hybrid vehicles with electric drivetrains. The major difference in electric vehicles is that their driving systems consist of a control system and motor system as well as energy storage devices such as batteries, a supercapacitor or flywheel [1]. The maximum speed that can be obtained for EVs strongly depends on the performance of the electric motor as well as the efficiency of the converters and their controller. In addition, the efficiency of the EVs depends on the differential, transmission components and the shaft.

One of the major components related to the power and consumption of vehicles is the transmission system. There are three basic types of vehicle transmission: automatic, manual and continuous variable [2]. The efficiency of the manual transmission is higher, but the automatic and continuous variable transmissions are more comfortable for usage. The transmission system is used for transmitting the torque from the electric motor to the wheels.

The hybrid electric vehicle combines different energy sources and transformations; this can involve an internal combustion engine equipped with an alternative environmentally friendly fuel, among other configurations. The interest herein is a hybrid vehicle in which one of the energy sources is electric.

The primary energy source for vehicles is the internal combustion engine (ICE), which has limited potential for the improvement of efficiency, including through application of high-pressure turbocharging, the Miller cycle, biofuels, new combustion processes such as HCCI, RCCI, etc., as well as the reduction of mechanical losses [3–6]. A significant step to improve vehicle efficiency is hybrid technology, which combines the advantages of the ICE and the electric propulsion. Hybrid electric vehicles (HEVs) will be a major part of the global vehicle fleet in years to come due to the fact that they offer $CO_2$ emission levels below the reference value for 2020 [6].

In order to assess BEV efficiency as well as to optimize parameters of the traction motor, battery and power converters, the power needed for propulsion and auxiliaries has to be estimated under real operating conditions [7–9]. However, the vehicle real operation is strongly unpredictable, depending on road conditions, weather, traffic, driver, etc. For this reason, power consumption is usually assessed over the standardized driving cycle imposed by legislation. In the EU, a new worldwide light-duty test cycle (WLTC) was adopted in 2017 as more representative of the homologation of new vehicles. Over the world, different test cycles are used, such as WLTC, NEDC, FTP, HWFET, USO6, etc. [10].

In [11], an EV simulation model was proposed in order to study the impact of ambient temperature on EV energy consumption and range over four test cycles (FUDS, NEDC, SFUDS and FIGE). Several simulation models of the same vehicle are presented in [12–15] using different approaches. The simulation model presented in [12] was developed taking into account a constant auxiliary consumption of 300 W and regenerative efficiency that depends on vehicle speed. The estimated energy consumption for UDDS and NEDC was 1.648 kWh and 1.547 kWh, respectively. The results from the model were compared to those obtained by means of a simulation software, FASTSim. Studies reported in [13,14] are related to the evaluation of the energy consumption over UDDS, HWFET and US06. The vehicle model was validated with experimental data published in the literature.

The BLDC electric machines most commonly used as a traction motor offer high maximum efficiency, within the range of 92% to 95% [11], as well as high vehicle driving dynamics and maximum speed without a complex transmission. However, in terms of on-board efficiency, more complex multiple speed transmission could be used. Spanoudakis et al. [12] studied the transmission effect on energy consumption of a small HEV developed for the Shell Eco-marathon competition. They revealed that a dual-speed gearbox could reduce the energy consumption up to 3.8%; in addition, the continuously variable transmission offered up to 4.3% lower energy consumption. [13–15] reported significant energy savings in a BEV (up to 14.1%) when using double ratio transmission and continuously variable transmission in comparison to single ratio transmission. The energy consumption evaluated within the HWFET, LA-92 and 4xECE driving cycle as double ratio transmission offered less energy consumption. Wu et al., in [16], studied the impact of a two-speed gearbox on EV energy consumption over FTP-75 and HWFET. The results revealed energy consumption improvement compared to a single-speed gearbox by 1.2% and 4.2%, respectively, for HWFET and FTP-75. The effect of a multispeed gearbox on energy consumption was studied in [16,17] over FTP-75 and NEDC test cycles. These studies revealed a potential to reduce the energy consumption up to 29% in the case of a three-speed gearbox. The energy consumption and cost optimization revealed that dual gear ratio transmission is the most promising solution for EVs. However, the results obtained over HWFET, FTP-75 and NEDC revealed the improvement variation within a wide range, from 1.2% to more than 14%. Moreover, the effect of the dual gear ratio transmission over WLTC (new homologation test cycle in EU) was not compared to the other cycles.

This modeling problem has multiple published results in worldwide articles, but the majority are based on the usage of specialized software [18–22], which is expensive and requires specific knowledge to employ. In practice, these models are oriented to specific types of architecture and to concrete components such as batteries, motors and supercapacitors. On the other hand, the basic correlations on which certain models of EVs are based are well known, and they are included in each model. The main task herein involves the creation of

a model that is implemented in one of the most popular environments for mathematical modeling—MATLAB/Simulink—in order to realize research goals based on defined driving cycles, to determine the energy needs, and on their basis to specify the requirements for the power electronic converters and the power of the control system, without the need for powerful computing resources and using standard computer configurations. On this basis, the concrete parameters of an EV can be determined using the components that build it, without addressing the concrete architecture.

The aim of this study is to evaluate the electrical energy needs to power an electric vehicle in the case of different transmission schemes (single and double speed gearbox) over the homologation cycles (WLTC, NEDC and FTP-75) by means of vehicle dynamics simulation. The study is limited to electric power for driving, without taking into consideration the energy needed to power the auxiliaries.

## 2. Mathematical Models

For the modeling of the vehicle, the following assumptions were made: the transmission efficiency is the same in one and two gears; the electronic converter controlling the electric motor ensures its operation in the most efficient zone; the driving style of the driver is not taken into account.

### 2.1. Vehicle Dynamics

In our study, a vehicle dynamic model was used, taking into consideration the traction force, aerodynamics and road resistance. The grade resistance was assumed to be zero due to the fact that in the simulation there is no slope of the road. Our dynamic model is shown in Figure 1.

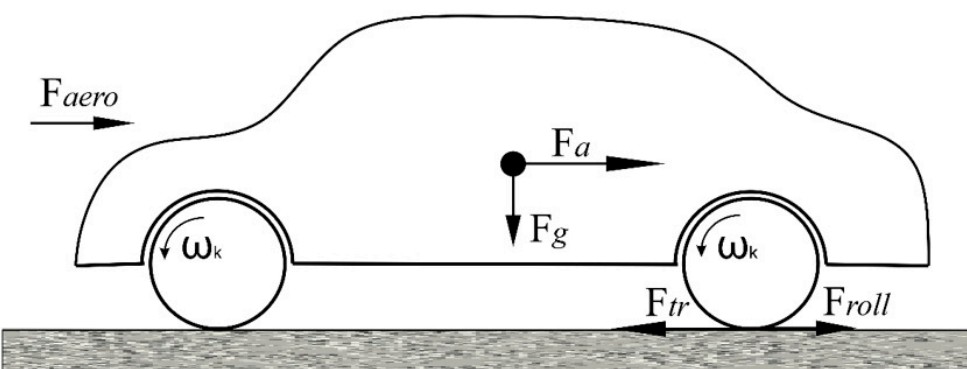

**Figure 1.** Vehicle dynamic model.

The acceleration of the vehicle depends on the forces applied to the vehicle and the equivalent vehicle mass. Thus, in our study the acceleration was estimated as follows:

$$\ddot{x} = \frac{F_{tr} - F_{aero} - F_{roll}}{m_{equiv}} \tag{1}$$

The traction force was estimated taking into account the traction motor output torque, transmission ratio, mechanical efficiency and wheel dynamic radius. Moreover, the maximum traction force was limited by the tire friction coefficient and the vertical load on the driving axle.

$$F_{tr} = \frac{M_{mot} \cdot i_{tr} \cdot \eta_{tr}}{r_d} \tag{2}$$

$$F_{tr(max)} = \mu_x \cdot F_z^{front} \tag{3}$$

The aerodynamic force was estimated taking into consideration the drag coefficient, vehicle frontal area, air density and vehicle speed, as follows:

$$F_{aero} = \frac{1}{2} \cdot C_d \cdot \rho_{air} \cdot S \cdot V^2 \tag{4}$$

The rolling resistance depends on tire design, road surface, vertical force and operating conditions such as tire pressure, temperature, etc. Thus, the rolling resistance force can be presented as a linear function of the vehicle speed using the following correlation:

$$F_{roll} = C_r + k \cdot V \tag{5}$$

where $C_r$ is the rolling friction coefficient and $k$ is the rolling speed coefficient. The equivalent vehicle mass is a sum of the vehicle total mass and an additional mass representing the inertial moment of rotational elements such as traction motor, transmission and wheels. Taking into account that the vehicle has a constant transmission ratio, it was assumed that the additional mass is proportional to the total vehicle mass. Then, the equivalent mass was estimated as follows:

$$m_{equiv} = C_i \cdot m \tag{6}$$

On the basis of vehicle dynamic model, the traction power and driving energy can also be estimated as follows:

$$P_{tr} = F_{tr} \cdot V \tag{7}$$

$$E_{tr} = \int_{t_1}^{t_2} P_{tr} \cdot dt \tag{8}$$

### 2.2. Traction Motor Model

In order to estimate the electrical consumption needed to drive the vehicle, a numerical model of traction motor losses was used. The model is based on the scaling motor losses presented in [23–28]. This approach assumes that the losses of any motor of the same type can be estimated by means of the loss coefficients of a motor with known losses. Then, the power loss is estimated by means of scaling the maximum speed and rated torque of the motors as follows:

$$P_{loss} = \left[ \frac{M_r \cdot \omega_{max}}{M_{ref} \cdot \omega_{ref}} \right] \cdot \left[ k_c \cdot M^2 \cdot \left( \frac{M_{ref}}{M_r} \right)^2 + k_i \cdot \omega \cdot \left( \frac{\omega_{ref}}{\omega_{max}} \right) + k_w \cdot \omega^3 \cdot \left( \frac{\omega_{ref}}{\omega_{max}} \right)^3 + C \right] \tag{9}$$

In the correlation presented above $M_{ref}$ and $\omega_{ref}$ are the rated torque and maximum speed, respectively, of the motor with existing power losses. The loss coefficients are estimated based on the Nissan Leaf motor, with a rated power of 80 kW. This motor offers maximum efficiency above 97% and inverter efficiency that reaches 99%. The parameters of the reference motor are listed in Table 1 [23].

**Table 1.** Reference traction motor data.

| Parameters | Designation | |
|---|---|---|
| Motor rated power (kW) | $P_{ref}$ | 80 |
| Motor rated torque (Nm) | $M_{ref}$ | 280 |
| Motor maximum speed (rpm) | $n_{ref}$ | 10,300 |
| Cooper losses coefficient (−) | $k_c$ | 0.14 |
| Iron losses coefficient (−) | $k_i$ | 2.3 |
| Windage losses coefficient (−) | $k_w$ | $6 \times 10^{-7}$ |
| Constant losses (−) | $C$ | 500 |

Then, the electric power to the traction motor and energy consumption were estimated by means of following correlations:

$$P_{el} = \frac{P_{tr}}{\eta_{tr}} + P_{loss} \tag{10}$$

$$E_{el} = \int_{t_1}^{t_2} P_{el} \cdot dt \tag{11}$$

### 2.3. Simulation Model

In order to estimate vehicle forces, a simulation model was developed in Matlab/Simulink. The main vehicle parameters that were used in the simulation are listed in Table 1. The model allows estimation of the instantaneous acceleration in case the traction force is defined. However, it can be used to estimate the traction force, power and energy when the instantaneous acceleration is known—the case for the test procedure (WLTP, NEDC, FTP, etc.). The first approach is used in order to validate the vehicle model; then, the second approach is used to simulate driving on the WLTP cycle.

In the case of maximum acceleration of the vehicle, the traction force is limited by the rated motor torque (Figure 2) as well as by the maximum traction force, depending on the slip conditions (3).

$$F_{tr} = \frac{M_r \cdot i_{tr} \cdot \eta_{tr}}{r_d} \leq \mu_x \cdot F_z^{front} \tag{12}$$

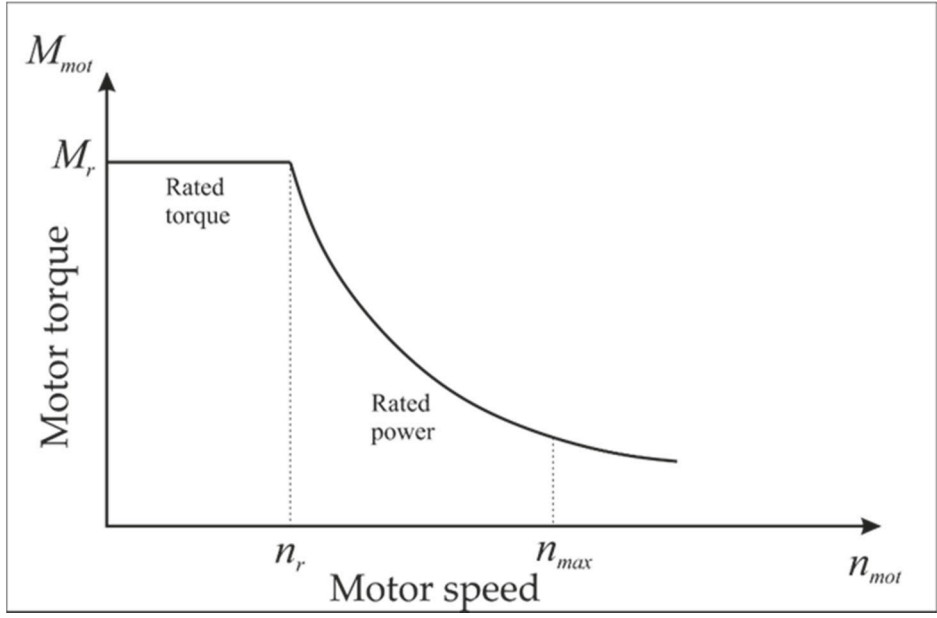

**Figure 2.** Motor output torque curve.

When $0 \leq n_{mot} \leq n_r$, the motor operates either at rated torque or at lower torque if the traction force is limited by the maximum longitudinal force in the tire contact with the road. When the motor speed is higher than $n_1$, the motor torque is additionally limited by the rated power, as follows:

$$M_{mot} = \frac{30 \cdot P_r}{\pi \cdot n_{mot}} \tag{13}$$

On the basis of the vehicle dynamics model (Equations (1)–(13)), a simulation model was developed in the MATLAB/Simulink environment. The structure of the model is presented in Figure 3. It allows estimation of instantaneous vehicle acceleration when the traction force is determined by means of acceleration pedal position. Therefore, this model was used only to validate the vehicle dynamic model. Then, based on the same equations,

a second numerical model was developed in the Matlab/Simulink environment, which allows estimation the vehicle power demand and energy consumption of driving on a test cycle [29–31]. However, in the second model, the vehicle instantaneous speed was an input parameter, determined by the cycle.

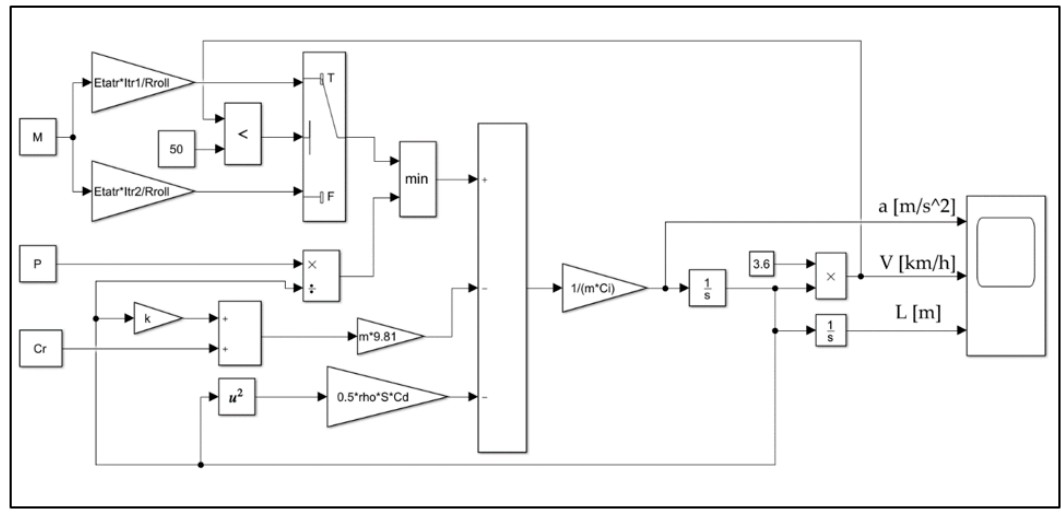

**Figure 3.** Block scheme of simulation model.

## 3. Model Validation

In order to validate our simulation model, an estimation of time for acceleration from 0 to 100 km/h was carried out. In this study, maximum traction force was applied to the wheels, limited by the motor rated torque, motor rated power and tire slip. It was considered an acceleration on a dry tarmac with friction coefficient $\mu_x = 0.8$ and the front axle vertical force of 50% of the total vertical force. The results concerning instantaneous acceleration vs. time and speed vs. time are presented in Figure 4.

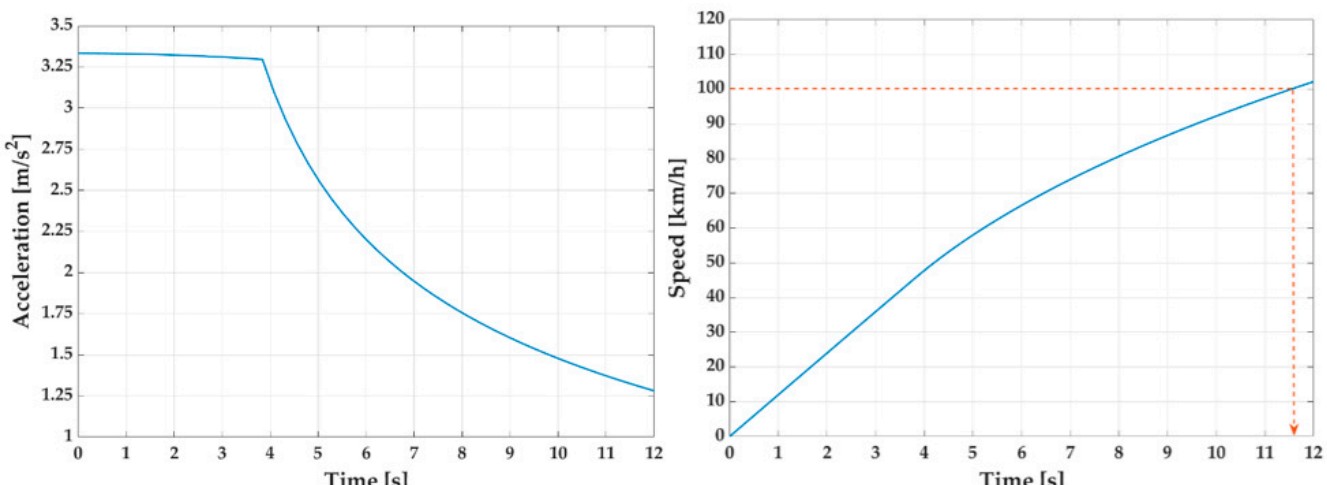

**Figure 4.** Vehicle acceleration vs. time and vehicle speed vs. time for acceleration from 0 to 100 km/h.

In order to achieve the same acceleration time from 0 to 100 km/h as vehicle catalog value (11.5 s), the mass correction coefficient $C_m$ was varied. The real value of this coefficient was unknown for us. Based on the simulation results, it was found that $C_m = 1.12$ provided an acceleration time of (11.53 s). Thus, this value was used further in our numerical study. The maximum instantaneous acceleration was estimated to be 3.33 m/s$^2$. It was limited by motor rated torque, as the maximum traction force was lower than the limited traction

force due to tire slip. The vehicle travel distance was also estimated for this acceleration and was determined as 188.7 m.

## 4. Simulation Results

Multiple examinations of the behavior of the different types of EVs were realized using several test procedures provided by Working Party on Pollution and Energy (GRPE). We studied the driving energy demand for the vehicle over WLTC, NEDC and FTP test cycles. These cycles are used for emissions testing as well as to test fuel consumption in homologation procedures. However, they represent the real vehicle behavior, taking into consideration urban, rural and highway driving conditions. Due to this, they could be used for the evaluation of energy demand as well as to estimate the autonomy of EVs. We aimed to assess the different driving dynamics of each cycle according to the specific energy consumption of an electric vehicle. WLTC is a part of the actual homologation procedure in the EU, as it replaced the NEDC in 2017. WLTC has three modifications, depending on the vehicle power to mass ratio. In this study, the WLTC representing Class 3 vehicles was used. The FTP-75 test procedure is adopted in the US. The main parameters of the cycles are listed in Table 2.

The speed variation versus time for each cycle concerned in this study is presented in Figure 5.

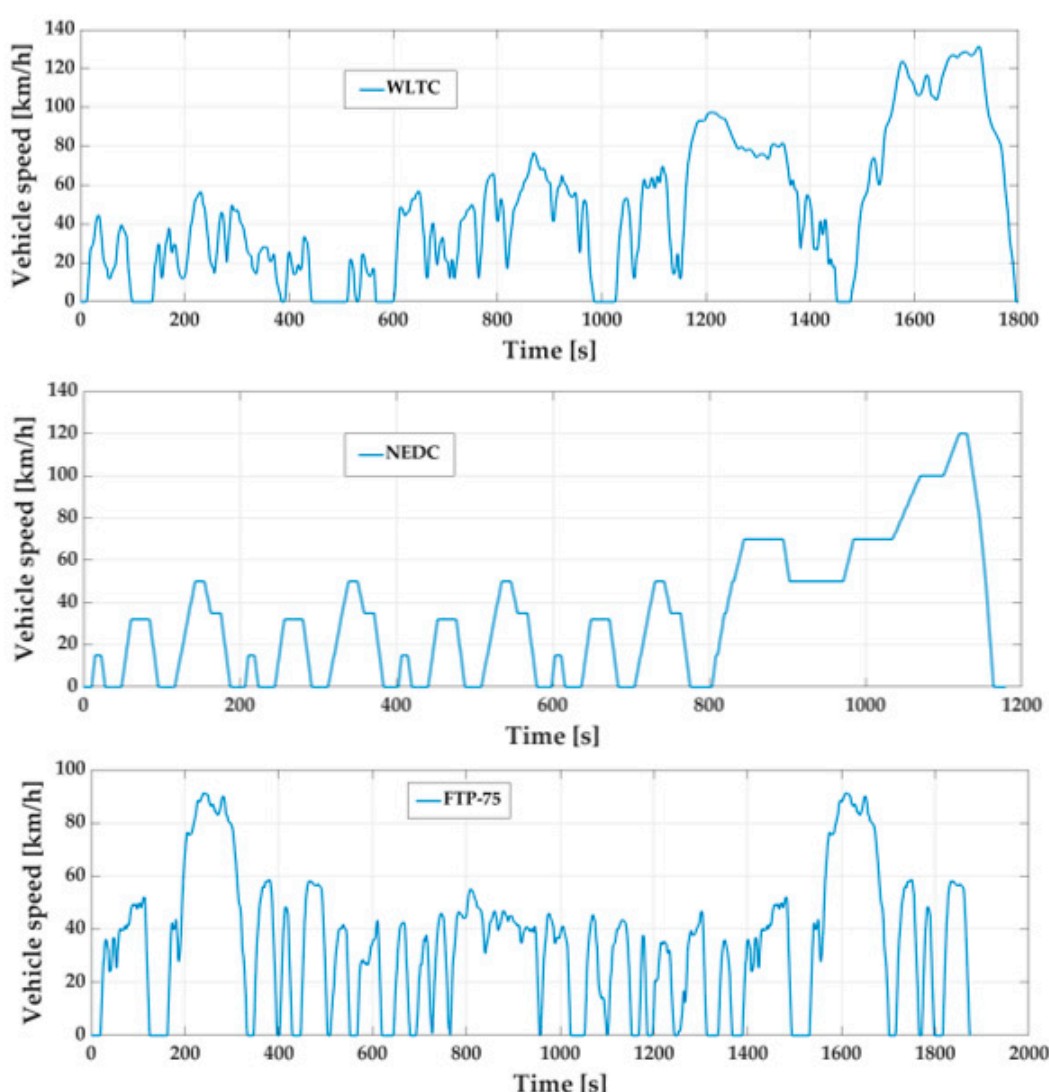

**Figure 5.** Vehicle speed vs. time for WLTC-Class 3, NEDC and FTP.

**Table 2.** Vehicle test cycles.

| Parameters | WLTC | NEDC | FTP-75 |
|---|---|---|---|
| Duration (s) | 1800 | 1180 | 1874 |
| Stops duration (s) | 235 | 241 | 241 |
| Distance (km) | 23.26 | 11.02 | 17.77 |
| Maximum speed (km/h) | 131.3 | 120 | 91.1 |
| Average speed (km/h) | 46.5 | 33.6 | 34.2 |
| Maximum acceleration (m/s$^2$) | 1.66 | 1.11 | 1.48 |
| Minimum acceleration (m/s$^2$) | −1.5 | −1.39 | −1.48 |

## 5. Numerical Study on Energy Consumption

### 5.1. Transmission Configuration and Gear Ratio

The vehicle driving energy depends on vehicle mass, geometric parameters, aerodynamic characteristics, road slope and instantaneous vehicle acceleration. When it is studied over a standardized test cycle, all of the parameters are usually defined either by the cycle or the vehicle data. Thus, for each vehicle, the driving energy depends mostly on the driving cycle dynamics. However, in terms of electric power supply and energy consumption to comply with driving energy demand, the motor/controller efficiency and transmission efficiency need to be taken into account. The transmission efficiency depends mostly on the torque; for the basic energy consumption evaluation, it could be assumed as a constant for each gear ratio. The motor/controller efficiency, however, depends on the motor operating point, which is related to vehicle speed and transmission ratio [23–26]. Thus, the transmission ratio could affect the motor efficiency at any given load and speed condition of the testing cycle.

In this study we aim to compare the energy consumption in the case of single and dual speed gearboxes. The transmission ratio for single speed transmission was taken from the prototype used in this study: 8.19. For dual speed transmission, the gear ratios for first and second gear as well as the gear switching speed was optimized in terms of the minimum energy consumption by the propulsion over WLTC. In this case, the first gear ratio was determined by means of maximum traction force that could be applied on the front axis. It was estimated by correlation (12), considering a weight distribution of 55% to the front axis. Thus, the first gear ratio was estimated to be 9.27.

The second gear ratio could vary over a wide range. The first restriction that needs to be taken into account is the maximum vehicle speed, estimated based on motor nominal electric power and maximum motor speed. For the studied vehicle, with the parameters defined in Table 3, the theoretical maximum speed was estimated to be 205 km/h. In order to achieve this speed at maximum motor rotational speed (10,000 rpm), a gear ratio of 5.79 was used. However, the motor offers constant output power within the range of 3000 to 10,000 rpm. Thus, any gear ratio within the range of 1.74 to 5.79 can provide the maximum speed of 205 km/h. Moreover, if the maximum speed is reduced bellow 205 km/h, a gear ratio up to 8 can be used. In our case, we carried out the simulation of electrical energy consumption when the second gear ratio varied within the range of 2 to 8 for gear switching speed from 40 to 70 km/h. The simulation was conducted over WLTC. The results are presented in Figure 6.

The result revealed that for each switching speed there is a gear ratio that offers minimum energy consumption by the driveline. However, this gear ratio depends on switching speed. We observed a low impact of gear ratio and switching speed for a ratio of more than 5. Low values of gear ratio, below 4, led to high impact on energy consumption, especially for a switching speed below 60 km/h. The combination of gear ratio and gear switching speed that offers minimum cumulative energy is 4.5 and 50 km/h. These values are further used in order to evaluate the dual speed transmission compared to single speed.

**Table 3.** Main vehicle data.

| Parameters | Designation | |
| --- | --- | --- |
| Type of EV | | BEV |
| Aerodynamic drag coefficient | $C_d$ | 0.28 |
| Frontal area (m$^2$) | $S$ | 2.19 |
| Rolling friction coefficient | $C_r$ | 0.0083 |
| Rolling speed coefficient | $k$ | $1 \times 10^{-5}$ |
| Vehicle mass (kg) | $m$ | 1645 |
| Mass correction coefficient | $C_m$ | 1.1 |
| Motor rated power (kW) | $P_r$ | 80 |
| Motor rated torque (Nm) | $M_r$ | 254 |
| Motor rated speed (rpm) | $n_r$ | 3000 |
| Motor maximum speed (rpm) | $n_{max}$ | 10,000 |
| Maximal speed (km/h) | | 144 |
| Acceleration time (s) | 0–100 km/h | 11.5 |
| Transmission ratio | $i_{tr}$ | 8.19 |
| Wheel rolling radius (m) | $r_{roll}$ | 0.315 |
| Transmission mechanical efficiency (-) | $\eta_{tr}$ | 0.95 |

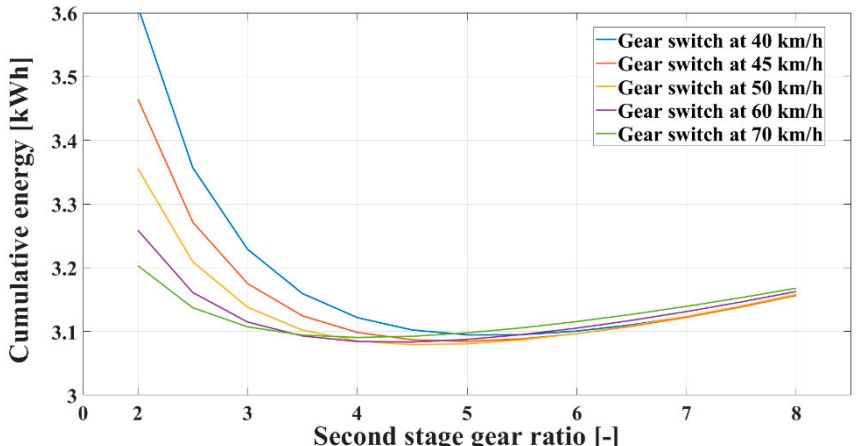

**Figure 6.** Cumulative energy consumption vs. transmission second gear ratio for different switching speed over WLTC.

It can be observed that in this case the single and dual speed transmission has nearly the same efficiency, since only gear can operate at a time. This phenomenon is addressed in the literature in multiple studies [12–16]. The appropriate operation of single and dual gearboxes can be achieved for electric vehicles. Thus, when using multiple speed transmission, the operating modes have approximately the same efficiency.

*5.2. Vehicle Power Demand*

A numerical study was carried out over the test cycles presented above, taking into account the motor efficiency. Thus, the vehicle mechanical power for driving as well as the motor electric power were estimated. The braking energy that could be recovered in the case of electric propulsion was also considered. During the braking phases, it was assumed that the vehicle can stop to zero only by means of the electric machine. However, the motor braking power was limited to 20 kW. For each driving cycle, two cases were studied. In the first case, a single speed gearbox was chosen, with a transmission ratio of 8.19. In the second case, a dual speed transmission ratio was studied, with a transmission ratio of 9.27 and 4.5 for the first and second gears, respectively. The gear is switched at 50 km/h. The instantaneous variation of the vehicle driving power and motor electric power were then estimated. The results concerning the single speed gearbox are presented in Figures 7–9. Power demands in the case of dual speed transmission are not presented on the graphs

due to the fact that the deviation in comparison to the single speed gearbox is invisible at this scale factor.

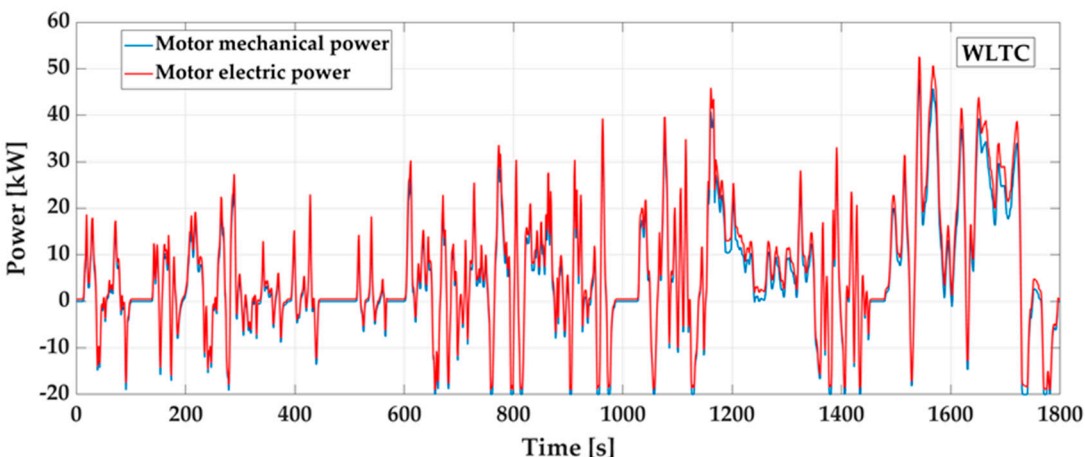

**Figure 7.** Motor electric and mechanical power vs. time for WLTC in the case of a single speed gearbox.

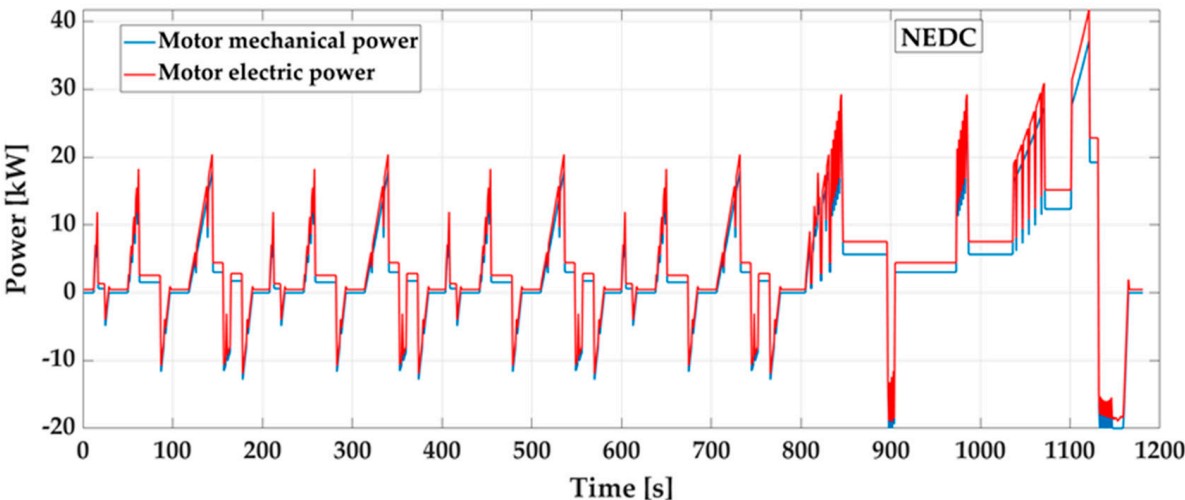

**Figure 8.** Motor electric and mechanical power vs. time for NEDC in the case of a single speed gearbox.

The simulation results revealed higher maximum electric power demand in WLTC. The maximum value for that cycle was estimated during the extra high part, as it was 52.6 and 54.5 kW for a single and double speed gearbox, respectively. In FTP-75, the maximum electric power demand was 47.4 and 51.2 kW, respectively. That values were estimated twice at the maximum vehicle speed in the cycle. The cycle that demanded the least power was NEDC. Here, the maximum electric driving power was observed to be 41.8 and 41.2 kW, respectively during the extra-urban diving part. The driving power variation over the cycle is important, due to fact that it defines the operating power of the traction motor, the inverter and the discharging of the battery.

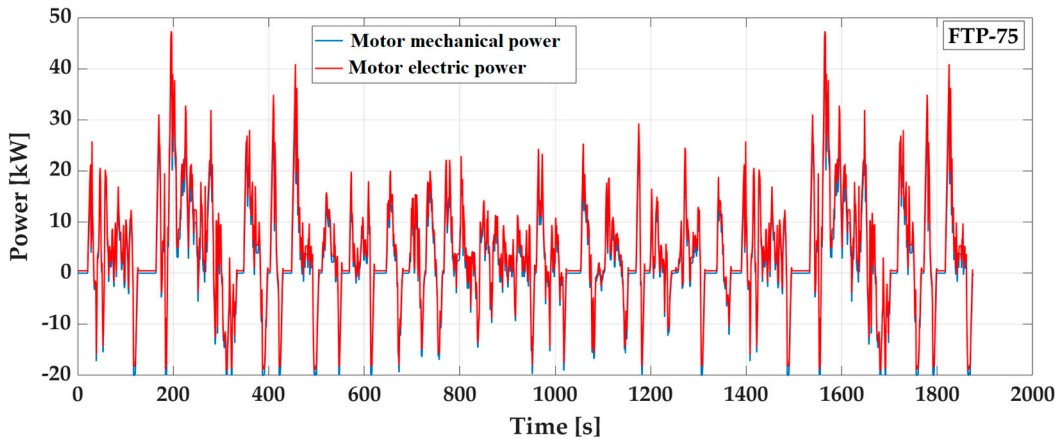

**Figure 9.** Motor electric and mechanical power vs. time for FTP-75 in the case of a single speed gearbox.

Different gear ratios lead to variation in the motor operating points, and thus variation in the motor efficiency occurs. An evaluation of the motor efficiency in the case of single and double speed gearboxes over the test cycles WLTC, NEDC and FTP-75 is presented in Figures 10–12.

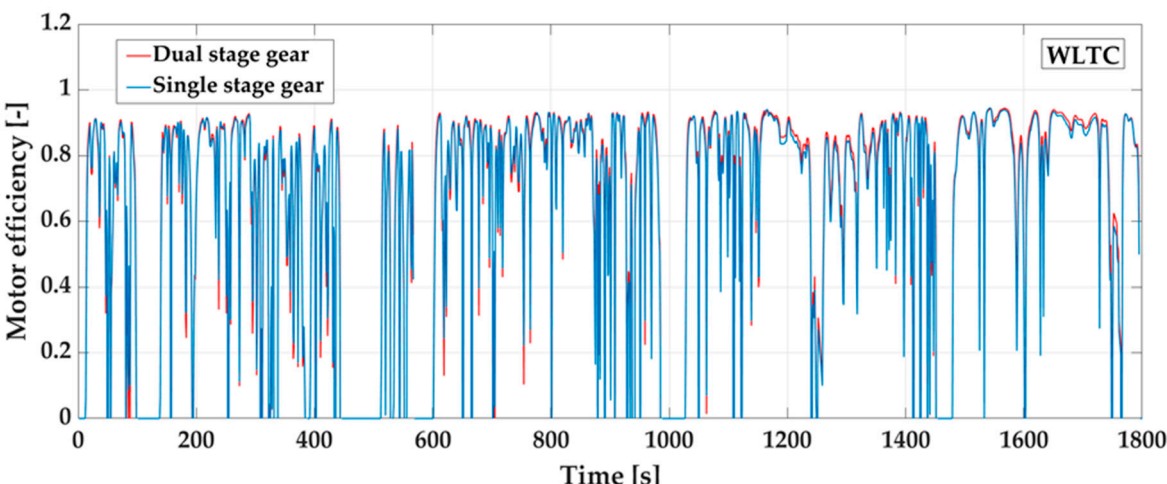

**Figure 10.** Motor efficiency vs. time for WLTC.

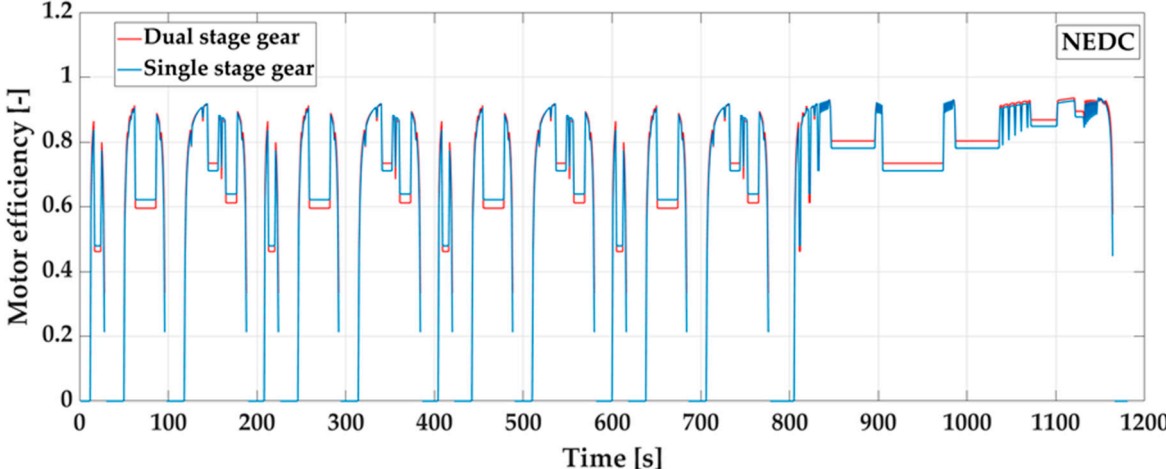

**Figure 11.** Motor efficiency vs. time for NEDC.

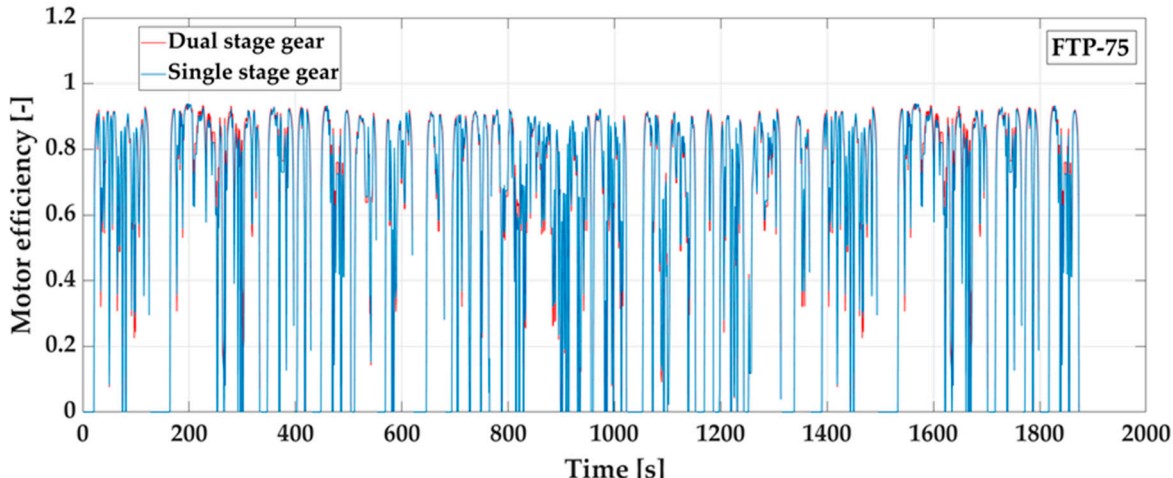

**Figure 12.** Motor efficiency vs. time for FTP-75.

In the case of single speed transmission, the maximum motor efficiency was found to be 95.2%, 95.3% and 95.3% in WLTC, NEDC and FTP-75, respectively. Very similar values were estimated for dual speed transmission, as they account to 95.4%, 95.2% and 95.4%, respectively, in WLTC, NEDC and FTP-75. However, the dual speed gearbox led to higher motor efficiency during the ''high'' and ''extra high'' part of WLTC as well as during the "extra-urban driving cycle"—the last part of NEDC. Less important variation in motor efficiency was observed in FTP-75. Single speed transmission offers better efficiency at some points at "low" and "medium" parts of WLTC and the "urban" part of NEDC.

### 5.3. Vehicle Energy Consumption

In order to evaluate the cycle impact on driving electric energy, an analysis of the cumulative energy was conducted for the cases explained above. The electric energy consumption was estimated by means of time integration of the instantaneous motor electric power. The results obtained for the studied cycles are presented in Figure 13.

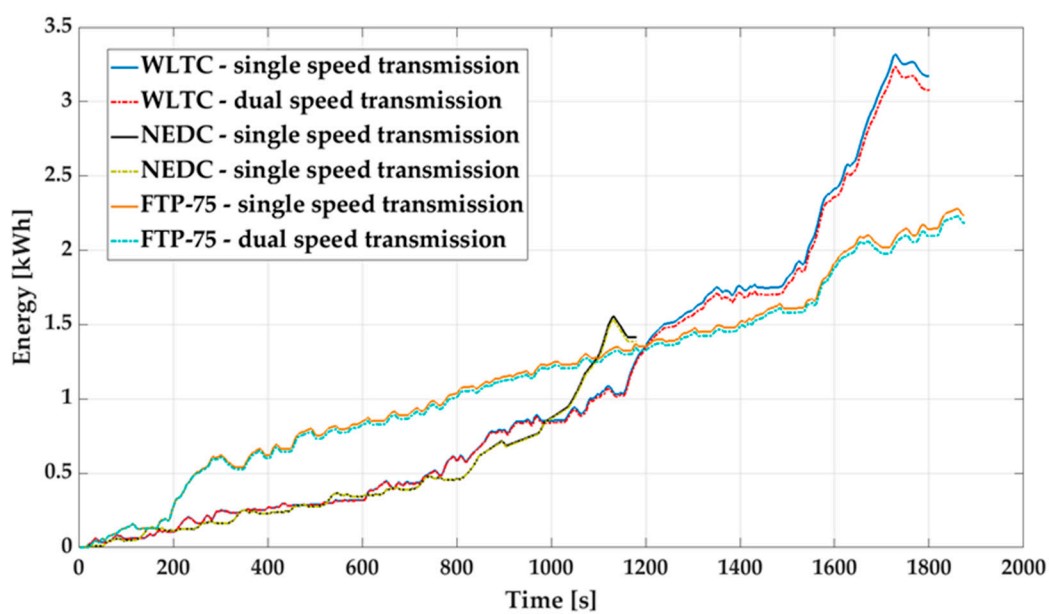

**Figure 13.** Cumulative electric energy vs. time for WLTC, NEDC and FTP-75.

The maximum cumulative energy was observed in WLTC, as the total electric consumption at the end of test cycle was 3.173 kWh for a single step gear and 3.079 kWh in the

case of a dual speed gearbox. In FTP-75, the total energy consumed was estimated to be 2.235 kWh and 2.184 kWh, respectively, for single and dual speed gearboxes. The lesser power demanded by NEDC led to an energy consumption of 1.415 kWh and 1.389 kWh, respectively, for single and dual speed gearboxes. The result obtained in NEDC with single speed transmission is similar to that estimated in [21]: 1.498 kWh obtained by means of FASTSim and 1.547 kWh obtained by means of the proposed model. Moreover, these values are estimated, taking into consideration the auxiliary power consumption of 300 W. In this case, the energy consumed by auxiliaries is 0.0983 kWh, and the driving electric energy is 1.4 kWh and 1.449, respectively, for model in FASTSim and the developed model. Based on cumulative energy, the cycle average power was also determined.

However, the analysis of the electric power demand and cumulative energy versus time do not provide enough information for energy consumption for driving over the standardized cycles due to different dynamics of the cycles. Therefore, the specific energy consumption based on the travelling distance was estimated. The results are listed in Table 4.

**Table 4.** Energy consumption and computational time.

| Parameters | WLTC | WLTC | NEDC | NEDC | FTP-75 | FTP-75 |
|---|---|---|---|---|---|---|
| | Single Speed Gearbox | Dual Speed Gearbox | Single Speed Gearbox | Dual Speed Gearbox | Single Speed Gearbox | Dual Speed Gearbox |
| Total energy per cycle (kWh) | 3.173 | 3.079 | 1.415 | 1.389 | 2.235 | 2.184 |
| Average power (kW) | 6.35 | 6.16 | 4.32 | 4.23 | 4.29 | 4.19 |
| Specific energy (kWh/km) | 0.136 | 0.132 | 0.129 | 0.126 | 0.126 | 0.123 |
| CPU time (ms) | 1.426 | 1.705 | 1.325 | 1.541 | 1.551 | 1.746 |

The average computational time for each simulation is also listed in Table 4. The simulation model was executed in the Matlab environment on a PC equipped with Intel(R) Core(TM) i-5-3330 CPU @ 3.00 GHz and 16 GB RAM. It was observed that the computational time depends on cycle duration and that the dual speed transmission model increased the CPU time needed. Due to very short computational time, the model can be further used for optimization of vehicle components such as gear ratio, traction motor, etc.

## 6. Conclusions

The impact of the transmission configuration in terms of gear ratio and gear numbers on the driving energy consumption was studied in this paper. The energy consumption in the case of single and dual speed transmission was evaluated over three test cycles (WLTC, NEDC and FTP-75), offering different maximal speed and vehicle dynamics.

A vehicle simulation model was developed and then applied to an EV. It was based on a vehicle dynamic model, taking into consideration vehicle driving resistances: aerodynamic, rolling and inertial force as well as the traction force. The scaling model of the traction motor was also implemented in the model in order to estimate the motor efficiency. The simulation model was developed in the MATLAB/Simulink environment. The model was first validated on the basis of vehicle acceleration time from 0 to 100 km/h on a dry tarmac surface. Then, a second computational program was used to simulate the vehicle energy consumption.

For single speed transmission, a gear ratio of 8.19 was chosen, the same as the prototype vehicle in the simulation. The dual speed transmission needed more complex optimization of the first and second gear ratio as well as the gear switch speed. The first gear ratio was estimated based on maximum traction force, while the second gear ratio and gear change speed were optimized in terms of minimum driving energy consumption over WLTC. Then, a first gear ratio of 9.27, a second gear ratio of 4.5 and gear change speed of 50 km/h were applied for other cycles.

The driving energy consumption was estimated for both single and dual speed transmission and compared over the test cycles. The results revealed that specific driving

energy per kilometer is similar for each test cycle. Despite very different driving dynamics between WLTC and NEDC, the specific energy for driving was 0.136 and 0.132 kWh/km in WLTC and 0.129 and 0.126 kwh/km in NEDC for single and dual speed transmission, respectively. It was found that the cycle that demanded the least energy is FTP-75 with a specific energy of 0.126 and 0.123 kWh/km, respectively. Concerning, the simulation results with single and dual speed transmission, it can be stated that a dual speed gearbox offers better energy efficiency, within the range of 2.3% to 2.9%. The higher value was observed on WLTC, while the lesser improvement was found over NEDC. The effect of the dual speed transmission can be increased if the motor braking power limitation is not included. In this case, the energy consumption for WLTC can be reduced by 3%. However, implementation of a dual speed gearbox increases the complexity of the transmission as well as increases the weight of the vehicle.

A developed and verified complex model of EVs is determined for their energy needs depending on the standard generally accepted for driving cycles and different topologies of vehicles. Based on these studies, the input parameters for hybrid energy sources such as supercapacitors, fuel cells and batteries were determined. In addition, this integrated model proposes a simplified design of power electronic systems in EVs, corresponding with their specific needs, including the choice of electric motor, the alternative of a power converter and the selection of appropriate devices for energy storage.

The model presented in this study can be further successfully applied to a more complex vehicle model and can consider the battery operating point and efficiency. Thus, a parametric optimization of EVs and HEVs could be carried out. Moreover, the model can be used to evaluate the recoverable braking energy.

**Author Contributions:** N.H., P.P., B.G. and G.V. were involved in the full process of producing this paper, including conceptualization, methodology, modeling, validation, visualization and preparing the manuscript. All authors have read and agreed to the published version of the manuscript.

**Funding:** This research was funded by: National Scientific Fund, grant number ДН07/06/15.12.2016 and European Regional Development Fund, grant number BG05M2OP001-1.001-0008 and the APC was funded by National Scientific Fund.

**Acknowledgments:** The research carry out is realized under the framework of the project "Model based design of power electronic devices with guaranteed parameters", ДН07/06/15.12.2016, Bulgarian National Scientific Fund, and was supported by the European Regional Development Fund within the Operational Programme "Science and Education for Smart Growth 2014–2020" under the Project CoE "National center of mechatronics and clean technologies "BG05M2OP001-1.001-0008.

**Conflicts of Interest:** The authors declare no conflict of interest.

## Nomenclature

| | |
|---|---|
| $\ddot{x}$ | Longitudinal vehicle acceleration |
| ICE | internal combustion engine |
| HEV | hybrid electric vehicle |
| BEV | battery electric vehicle |
| EV | electric vehicle |
| $F_{tr}$ | traction force |
| $F_{aero}$ | aerodynamic force |
| $F_{roll}$ | rolling resistance |
| $m_{equiv}$ | equivalent vehicle mass |
| $M_{mot}$ | traction motor output torque |
| $i_{tr}$ | transmission ratio |
| $\eta_{tr}$ | mechanical efficiency |
| $r_d$ | wheel dynamic radius |
| $F_{tr(max)}$ | maximum traction force |
| $\mu_x$ | tire friction coefficient |
| $F_z^{front}$ | vertical load on the driving axle |

| $C_d$ | drag coefficient |
|---|---|
| $S$ | vehicle frontal area |
| $\rho_{air}$ | air density |
| $V$ | vehicle speed |
| $C_r$ | rolling friction coefficients |
| $k$ | rolling speed coefficient |
| $m$ | vehicle mass |
| $P_{tr}$ | traction power |
| $E_{tr}$ | driving energy |
| $P_{loss}$ | power loss |
| $M_{ref}$ | rated torque |
| $\omega_{ref}$ | maximum speed of the motor |
| $P_{el}$ | electric power |
| $E_{el}$ | energy consumption |
| $n_{mot}$ | motor speed |
| $C_m$ | mass correction coefficient |

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
