# Peer review of "Model-Based Estimation of Transmission Gear Ratio for Driving Energy Consumption of an EV"

_electronics, doi:10.3390/electronics10131530_

Round 1
Reviewer 1 Report
Dear authors,
I appreciate your great research potential in the field and the substantial efforts involved in this research activity, which is very attractive to a large community of readers and specialists working in the field of HEV / EV. To improve the overall quality and also to increase the visibility of your research, I would like to make the following comments and suggestions:
-
For a better reading, please insert in the manuscript a table with all the acronyms and abbreviations used in the text.
-
In the introductory section, if necessary, I think it is very useful to compare the results of this paper with those obtained in the field of literature.
-
In the reference section, please add at least 2-3 of the most recent works in the field of literature (ie those published in 2020 and 2021)
-
I think that it could also be useful to formulate the optimization problem for the minimum driving power consumption on standardized test cycles is useful for all implementers working in the MATLAB / Simulink environment.
-
A study of the impact of uncertainties and disturbances that could be added in the dynamic model on the accuracy of the dynamic model would also be very attractive.
Author Response
First of all, we would like to thank you for the thorough review of our paper (electronics- 1254330) and the useful remarks to improve it.
Reviewer 1
Comments to the Authors
Dear authors,
I appreciate your great research potential in the field and the substantial efforts involved in this research activity, which is very attractive to a large community of readers and specialists working in the field of HEV / EV. To improve the overall quality and also to increase the visibility of your research, I would like to make the following comments and suggestions:
- For a better reading, please insert in the manuscript a table with all the acronyms and abbreviations used in the text.
- In the introductory section, if necessary, I think it is very useful to compare the results of this paper with those obtained in the field of literature.
- In the reference section, please add at least 2-3 of the most recent works in the field of literature (ie those published in 2020 and 2021)
- I think that it could also be useful to formulate the optimization problem for the minimum driving power consumption on standardized test cycles is useful for all implementers working in the MATLAB / Simulink environment.
- A study of the impact of uncertainties and disturbances that could be added in the dynamic model on the accuracy of the dynamic model would also be very attractive.
To Reviewer 1:
Thank you for your review and valuable remarks.
- For a better reading, please insert in the manuscript a table with all the acronyms and abbreviations used in the text.
-Thank you for your remark. Really, the manuscript will be more readable if we add an acronyms and abbreviations. We added in the end of this research.
- In the introductory section, if necessary, I think it is very useful to compare the results of this paper with those obtained in the field of literature.
- Thank you for your remark. We made some changes in the introduction part and we added different studied models.
- In the reference section, please add at least 2-3 of the most recent works in the field of literature (ie those published in 2020 and 2021)
- Thank you for your remark. We also think that will be valuable to have the newest research in this area. For this reason we added some new references in the bibliography.
- I think that it could also be useful to formulate the optimization problem for the minimum driving power consumption on standardized test cycles is useful for all implementers working in the MATLAB / Simulink environment.
- Thank you for your remark. For further researches in this area, we propose on the base of the presented model of electric vehicle to solve an optimization problem for choosing an appropriate topology of the power electric drive. Also, with detailed researches to realize a determination of accurate values of the gear box and the switching speed.
- A study of the impact of uncertainties and disturbances that could be added in the dynamic model on the accuracy of the dynamic model would also be very attractive.
- Thank you for your remark. In this moment we don’t consider the impact of the uncertainties and the disturbances. It will be helpful for further researches to realized a study for all the uncertainties and disturbances such as the behavior of the driver, the traffic in the city and etc. In this case it will be helpful to used an advanced methodologies and approaches for control based on artificial intelligence such as neuron network and model-based optimization.
Thank you very much for the exact review!
Reviewer 2 Report
In my opinion, the work is interesting.
Since there are several works concerning proposed optimizations authors should focus their attention on innovation content and in the way in which they are able to emphasize it especially respect recent works.
Also, concern the development of electric vehicle models with various subsystems.
Also I have a doubt concerning the modelling of efficiencies since in electric vehicles motors operate in multiple quadrant efficiency in braking is different respect to traction. Also the way in which losses modify performance is different and this consideration is not so clear in this work.
Finally for what concern adopted cycles some further discussion has to be performed respect to the realims of same cycles especially NEDC since in more recent works this choice is often discussed
Author Response
Reviewer 2
First of all, we would like to thank you for the thorough review of our paper (electronics- 1254330) and the useful remarks to improve it.
Comments to the Authors
In my opinion, the work is interesting.
- Since there are several works concerning proposed optimizations authors should focus their attention on innovation content and in the way in which they are able to emphasize it especially respect recent works.
- Also, concern the development of electric vehicle models with various subsystems.
- Also I have a doubt concerning the modelling of efficiencies since in electric vehicles motors operate in multiple quadrant efficiency in braking is different respect to traction. Also the way in which losses modify performance is different and this consideration is not so clear in this work.
- Finally for what concern adopted cycles some further discussion has to be performed respect to the realims of same cycles especially NEDC since in more recent works this choice is often discussed
To Reviewer 2:
Thank you for your review and valuable remarks.
- Since there are several works concerning proposed optimizations authors should focus their attention on innovation content and in the way in which they are able to emphasize it especially respect recent works.
- Thank you for your remark. After your review we are added several recent researches and they are described in the introduction of our work.
- Also, concern the development of electric vehicle models with various subsystems.
- Thank you for your remark. For further researches related to this problem the main purpose will be the received now results to be exposed on specific architecture because on the base of the obtained consumption of the power and energy we will choose an optimal architecture.
- Also I have a doubt concerning the modelling of efficiencies since in electric vehicles motors operate in multiple quadrant efficiency in braking is different respect to traction. Also the way in which losses modify performance is different and this consideration is not so clear in this work.
- Thank you for your remark. The switching of gearbox is in the efficiency area. In this case we provide that control of the power electronic converters will ensure the necessary efficiency operation mode.
- Finally for what concern adopted cycles some further discussion has to be performed respect to the realims of same cycles especially NEDC since in more recent works this choice is often discussed
- We are agreed with your recommendation. In our research we study several driving cycles and we observed that the total energy of the proposed driving cycles is nearly the same. The differences can be observed of the maximum energy value. From this point of view we can defined the variables in the sybsystems. On the other hand, the chosen approach in modeling makes it possible to simulate any cycle of motion for a long enough time - on the order of tens of minutes, without the need for high computing power.
Thank you very much for the exact review!
Reviewer 3 Report
This manuscript presents a dual gear ratio transmission in EV, but it doesn’t provide several important aspects such as transmission shift logic. Also, the manuscript assumes the dual gear transmission has same efficiency as single gear transmission, which is questionable. Normally when more mechanical parts are added, efficiency decreases. Thus, the simulation results and the conclusions of this manuscript are doubtable. Work on EV with dual gear transmission has been conducted by other researchers and published previously. This manuscript doesn’t show much improvements on modeling methods or transmission design/control.
The following questions need to be addressed.
- Mathematical Models on Pages 3-7: the manuscript doesn’t include a driver model. Do you have a driver model? If yes, please add some description.
- Transmission Configuration and Gear Ration on Page 9: please provide an introduction to the mechanical structure of the dual gear ratio transmission.
- Lines 296-297: The manuscript makes an unrealistic assumption that the transmission efficiency is a constant for different gear ratios. The transmission efficiency should be different between dual gear transmission and single gear transmission. Normally when more mechanical parts are added, efficiency decreases. If the authors assume it is a constant, please add some explanation.
- Page 10: what is the transmission shift algorithm? Please add some explanation. Does it shift by speed or power command?
- Figure 9 on Page 12, label of “Pel……” is incomplete.
Author Response
Reviewer 3
First of all, we would like to thank you for the thorough review of our paper (electronics- 1254330) and the useful remarks to improve it.
Comments to the Authors
This manuscript presents a dual gear ratio transmission in EV, but it doesn’t provide several important aspects such as transmission shift logic. Also, the manuscript assumes the dual gear transmission has same efficiency as single gear transmission, which is questionable. Normally when more mechanical parts are added, efficiency decreases. Thus, the simulation results and the conclusions of this manuscript are doubtable. Work on EV with dual gear transmission has been conducted by other researchers and published previously. This manuscript doesn’t show much improvements on modeling methods or transmission design/control.
The following questions need to be addressed.
- Mathematical Models on Pages 3-7: the manuscript doesn’t include a driver model. Do you have a driver model? If yes, please add some description.
- Transmission Configuration and Gear Ration on Page 9: please provide an introduction to the mechanical structure of the dual gear ratio transmission.
- Lines 296-297: The manuscript makes an unrealistic assumption that the transmission efficiency is a constant for different gear ratios. The transmission efficiency should be different between dual gear transmission and single gear transmission. Normally when more mechanical parts are added, efficiency decreases. If the authors assume it is a constant, please add some explanation.
- Page 10: what is the transmission shift algorithm? Please add some explanation. Does it shift by speed or power command?
- Figure 9 on Page 12, label of “Pel……” is incomplete.
To Reviewer 3:
Thank you for your review and valuable remarks.
- Mathematical Models on Pages 3-7: the manuscript doesn’t include a driver model. Do you have a driver model? If yes, please add some description.
- Thank you for your remark. In this research the driver model is not included but this a very useful offer and can be discussed in future publication.
- Transmission Configuration and Gear Ration on Page 9: please provide an introduction to the mechanical structure of the dual gear ratio transmission.
- Thank you for your remark. In this research we consider the gearbox only as a ratio. In this case is not our purpose to study the mechanical structure. This is presented in several studies which is added in the references.
- Lines 296-297: The manuscript makes an unrealistic assumption that the transmission efficiency is a constant for different gear ratios. The transmission efficiency should be different between dual gear transmission and single gear transmission. Normally when more mechanical parts are added, efficiency decreases. If the authors assume it is a constant, please add some explanation.
- Thank you for your remark. In the section “Mathematical model” we are added the assumption at which we are realized our models. One of them is that the coefficient of transmission efficiency is a constant at different gear ratio. Reason for that is presented in multiple researches where are described similar problem. In our case at one moment, we have only one gear ration transmission.
- Page 10: what is the transmission shift algorithm? Please add some explanation. Does it shift by speed or power command?
- Thank you for your remark. The transmission shift algorithm is realized by speed command. The main purpose of this study is to determine the speed which is appropriate for shifting the transmission ratio for obtain the minimal consumption.
- Figure 9 on Page 12, label of “Pel……” is incomplete.
- Thank you for your detailed review. It is made a technical error and now it is corrected.
Thank you very much for the exact review!
Round 2
Reviewer 1 Report
Dear authors,
Thank you very much for your great efforts to give the right answers to all my comments. I agree also that in your future research work you could take into account the requirements from comment 4 and solve an optimization problem for choosing an appropriate topology of the power electric drive. For sure, I agree that a rigorous study on the impact of the model uncertainties and disturbances on the dynamic model accuracy, suggested in comment 5, can be made in your future work by developing advanced methodologies and approaches for control based on artificial intelligence.
Reviewer 2 Report
In my opinion it's a good paper and it can be published if possible it's should interesting to integrate the work with recent contribuitions on similar topics